# Spatial interpolation of health and demographic variables: Predicting malaria indicators with and without covariates

Camille Morlighem[1,2]*, Chibuzor Christopher Nnanatu[3,4], Corentin Visée[1,2], Atoumane Fall[5], Catherine Linard[1,2,6]

1 Department of Geography, University of Namur, Namur, Belgium, 2 ILEE, University of Namur, Namur, Belgium, 3 WorldPop, School of Geography and Environmental Science, University of Southampton, Southampton, United Kingdom, 4 Department of Statistics, Nnamdi Azikiwe University, Awka, Nigeria, 5 Agence Nationale de la Statistique et de la Démographie (ANSD), Dakar, Senegal, 6 NARILIS, University of Namur, Namur, Belgium

* camille.morlighem@outlook.be

## Abstract

Accurate mapping and disaggregation of key health and demographic risk factors have become increasingly important for disease surveillance, which can reveal geographical social inequalities for improved health interventions and for monitoring progress on relevant Sustainable Development Goals (SDGs). Household surveys like the Demographic and Health Surveys have been widely used as a proxy for mapping SDG-related household characteristics. However, there is no consensus on the workflow to be used, and different methods have been implemented with varying complexities. This study aims to compare multiple modelling frameworks to model indicators of human vulnerability to malaria (SDG Target 3.3) in Senegal. These indicators were categorised into socioeconomic (e.g., stunting prevalence, wealth index) and malaria prevention indicators (e.g., indoor residual spraying, insecticide-treated net ownership). We compared three categories of the commonly used methods: (1) spatial interpolation methods (i.e., inverse distance weighting, thin plate splines, kriging), (2) ensemble methods (i.e., random forest), and (3) Bayesian geostatistical models. Most indicators could be modelled with medium to high predictive accuracy, with $R^2$ values ranging from 0.40 to 0.86. No method or method category emerged as the best, but performance varied widely. Overall, socioeconomic indicators were generally better predicted by covariate-based models (e.g., random forest and Bayesian models), while methods using spatial autocorrelation alone (e.g., thin plate splines) performed better for variables with heterogeneous spatial structure, such as ethnicity and malaria prevention indicators. Increasing the complexity of the models did not always improve predictive performance, e.g., thin plate splines sometimes outperformed random forest or Bayesian geostatistical models. Beyond performance, we compared the different methods using other criteria (e.g., the ability to constrain the

**Data availability statement:** This study uses data from the 2017 Senegal Continuous DHS (with questions on population and housing, respondent's characteristics, nutrition of children and adults, malaria and water, sanitation and hygiene). DHS datasets are publicly available by registration and request to the DHS Program (https://dhsprogram.com/data/dataset_admin/login_main.cfm). The Guide to DHS statistics (https://dhsprogram.com/data/Guide-to-DHS-Statistics/) as well as the 2017 Senegal DHS report can be used to calculate the indicators of this study. National boundary shapefiles can be downloaded from the GADM website (https://gadm.org/). R scripts used to conduct this study are available at https://doi.org/10.6084/m9.figshare.24874218.v1. A simulated DHS dataset is provided at this link to demonstrate how the code works. All other data (covariates, interpolated surfaces, prediction grid) are available at the link above.

**Funding:** CM is a Research Fellow from the Fonds de la Recherche Scientifique (F.R.S.-FNRS) (https://www.frs-fnrs.be/fr/). The funder did not play any role in the study design, data collection and analysis, decision to publish, or preparation of the manuscript.

**Competing interests:** The authors have declared that no competing interests exist.

prediction range or to quantify prediction uncertainty) and discussed their implications for selecting a modelling approach tailored to the needs of the end user.

## Introduction

The Sustainable Development Goals (SDGs) were launched by the United Nations as a set of 17 goals and 169 related targets to be achieved through a collective effort by 2030, addressing the global challenges of social and economic inequalities, climate change and planet degradation. These goals include ending poverty and hunger, providing quality education and health for all, and achieving gender equality, among others [1]. The SDGs were defined with the aim of 'leaving no one behind', to ensure that everyone is included in this 2030 agenda [1]. Achieving these goals requires consistent tracking of progress towards the SDGs, and to this end, a series of 232 SDG indicators have been defined for evaluation and monitoring [1]. Yet, global and regional indicators mask large disparities at the sub-regional level, but also at finer scales, arising from complex social and environmental processes [2]. In this context, mapping these indicators helps to reveal spatial inequalities at different scales below the sub-regional level (e.g., urban/rural, intra-urban), thus contributing to 'leaving no one behind' [3]. Maps of SDG indicators at a finer scale can serve several purposes: they can help track the progress towards the SDG attainment, support decision-making to achieve the SDG agenda [3,4], help evaluate the impact of interventions and health investments [5], and can also be used as base covariate layers for spatial models of other health and demographic variables [4], e.g., malaria risk models [6,7].

Common approaches to mapping SDG indicators (e.g., poverty mapping) often rely on census data and geospatial data analysis, sometimes combined with ancillary survey data, such as small area estimation [3,8]. However, censuses are typically conducted every 10 years, if not more in some low-income countries, while monitoring SDG progress requires updated maps [3]. Other drawbacks include the unreliability and/or unavailability of census data and a coarse spatial resolution in some resource-constrained settings [4]. With these limitations, it is challenging for many government agencies to use census data to effectively monitor and assess SDG progress and plan interventions [4].

Since the mid-2000s, household survey programs such as the Demographic and Health Surveys (DHS), Malaria Indicator Surveys (MIS) and Multiple Indicator Cluster Surveys (MICS) come as a substitute for census data, especially in countries where the last census is outdated [3,5]. In particular, DHS provide estimates of key health and demographic indicators, based on nationally representative samples, that can help measure progress towards various SDGs: percent distribution of population by wealth quintiles (SDG 1: 'No poverty'), prevalence of stunting and anemia (SDG 2: 'Zero hunger'), HIV and malaria prevalence (SDG 3: 'Good health and well-being'), literacy rate (SDG 4: 'Quality education'), and percentage use of improved sanitation (SDG 6: 'Clean water and sanitation'), among others. These indicators can be

aggregated at the survey cluster level, for which geographic coordinates are available, and these can be interpolated to create continuous surfaces [9]. These maps help assess the progress towards the SDGs, as DHS are consistently conducted in more than 90 developing countries every 3–5 years [10] – or almost annually in some countries (e.g., Senegal). DHS hence provides an opportunity to map SDG indicators more accurately, at a higher spatial resolution, with regular updates in many countries [4]. Overall, DHS and MIS (from the same DHS program) have already been used for various mapping applications: mapping malaria prevalence [11–14], vaccination coverage [15,16], HIV prevalence [17], poverty [8,18,19], population age structure [4], ethnicity [20], female genital mutilation prevalence [21–23], etc.

However, although the mapping of DHS indicators is a widely studied topic, there is no consensus on the modelling workflow to be used for this purpose. Different types of models have been implemented with different levels of complexity and model inputs. Many of these studies used covariate-based approaches such as Bayesian geostatistical models [4,11,12,15,18] or machine learning techniques (e.g., random forest modelling, boosted regression trees) [8,13,19]. Yet, other research has shown that spatial interpolation methods that rely on recovering the spatial autocorrelation pattern in the data can also be effective for mapping DHS indicators (e.g., inverse distance weighting, splines) [17,20]. Given the wide variety of modelling approaches, some studies have compared Bayesian geostatistical models with other covariate-based methods: machine learning models [3,16], multivariate regression [24,25] and generalised additive models using spline interpolation [25]. [26] further compared several methods that use only spatial autocorrelation, including Bayesian geostatistical models, kriging and spatial random forest. Nonetheless, there has been no systematic comparison of all these approaches that use spatial covariates and/or spatial autocorrelation, and it is still unclear how all these methods will perform for different indicators. Although the DHS program recommends the use of model-based geostatistics with covariates [27], from a policy perspective, it is important to compare their added value with other approaches. For example, spatial interpolation methods that do not rely on ancillary covariate data can save significant time and computational effort in data processing and model implementation, compared to complex methods that use covariates or require higher computational power.

Among the SDGs, SDG Target 3.3 focuses on ending epidemics of communicable diseases, including malaria, which accounted for 249 million cases in 2022 [28]. In this context, malaria risk maps support policymakers in targeting control interventions and achieving the SDG target. However, these maps are often based solely on the hazards that influence the suitability of the environment for malaria mosquito vectors, such as climate and land cover variables, and rarely consider the vulnerability of society to malaria. Yet, socioeconomic factors and human behaviour regarding the use of preventive measures are known to influence malaria risk [29]. In general, the risk of malaria is known to be higher in areas with higher levels of poverty, as people may be more exposed to malaria disease, for example by living in houses with poor housing materials [30]. Malaria prevention measures such as insecticide-treated nets and indoor residual spraying have a direct impact on people's ability to anticipate malaria [29]. Women's education can also play an important role in prevention, as it may increase household income through better jobs and improve mothers' knowledge of malaria prevention [29,31]. Ethno-religious beliefs may also influence perceptions of malaria and the uptake of preventive measures [32]. Some people may also be more biologically susceptible to malaria, such as stunted children [29,33] or people suffering from malaria co-infections (e.g., schistosomiasis [34]). While hazard-related variables are more readily available from earth observation data, household surveys such as the DHS have the potential to provide such malaria vulnerability indicators, which can be mapped into continuous surfaces. Beyond malaria, mapping vulnerability indicators from DHS may also be relevant to other vector-borne diseases for which epidemiological survey data are not always available, such as dengue, and where maps of vulnerability indicators can assist in planning health interventions.

The aim of this paper is to produce continuous surfaces of useful malaria-related indicators from the DHS with the following sub-objectives: 1) compare three categories of methods for predicting DHS indicators (spatial interpolation methods, ensemble methods and Bayesian geostatistical models), 2) assess the added value of covariate-based methods over methods that rely only on spatial autocorrelation, and 3) provide a comprehensive assessment of the

strengths and weaknesses of these methods to guide users in their choice. In addition, the codes to implement these methods are also provided (see [35]). We used DHS indicators that drive malaria vulnerability: socioeconomic and malaria prevention indicators. This study focuses on Senegal, which has set a goal of eliminating malaria by 2030 [36]. Such interpolated surfaces of malaria-related indicators could (1) be used at a later stage to build spatially integrated malaria risk models, and (2) be useful for the planning of health interventions in Senegal to achieve the 2030 elimination target. In addition, the high availability of DHS data provides an opportunity for a follow-up study in Senegal or replication in other countries.

## Materials and methods

### DHS data and indicators

The DHS program provides cross-sectional estimates of demographic and health indicators sampled at the national level from over 400 surveys in more than 90 low- and middle-income countries. The DHS sampling frame is stratified by geographic region and rural/urban areas within each region. Across each stratum, primary sampling units (PSU), or clusters, are defined using enumeration areas (EAs) provided by the most recent national population census. PSU are selected with a probability proportional to their population size, and a group of households (around 25–30) is further selected within each PSU for questionnaire interview [10,37]. Geolocations for cluster centroids are made available along with the DHS recode files, but their geographic coordinates are randomly displaced up to 2 km in urban areas and 5 km in rural areas (with an additional 1% offset up to 10 km) to protect the privacy of the survey participants [37,38]. While rural areas are on average less affected by this displacement, previous works showed that it significantly affects the accuracy of spatial models of DHS indicators at the urban scale [8,9,14]. We further account for this displacement at the covariate extraction stage.

In this study, we used data from the Continuous DHS conducted in Senegal in 2017 by the National Agency of Statistics and Demography of Senegal (ANSD). This survey was chosen for two main reasons. First, it is the most recent household survey with the largest number of clusters for Senegal (i.e., 400), and second it was conducted mostly during the wet season (i.e., which lasts from June to October) [39]. This ensures that indicators related to malaria prevention are measured during the season of high malaria transmission. Based on DHS recommendations [27], we selected several indicators for spatial interpolation at the national level for Senegal. These fall into two categories: socioeconomic indicators and malaria prevention indicators. These indicators were aggregated at the cluster level following instructions from the Senegal DHS 2017 report [39] and general guidance provided by the DHS program [10]. They are summarized in Table 1 and described in more detail in S1 Text.

### Geospatial covariates

In this study, we compiled a set of open-source environmental and socioeconomic covariates that have been shown to correlate with DHS indicators in previous work [3,9,15]. However, in contrast to existing studies, we assembled datasets with higher spatial resolution (1 km at the coarsest) to improve the accuracy of predictions at the urban scale, as recommended in [3,8]. We selected covariates that matched 2017 as closely as possible, i.e., consistent with the DHS data. All the covariates collected in this study are listed with their characteristics in Table 2 and are described in more detail in S1 Text. Due to differences in spatial resolution, projection and extent, all covariates were resampled to a common 1x1 km grid resolution and continuous covariates were converted to z-scores to account for different units of measurement. Covariates were extracted in 5 km buffers around rural DHS cluster centroids and 2 km buffers around urban clusters, following DHS recommendations [43]. For continuous covariates, we extracted the average value per buffer. For categorical covariates, we extracted the average proportion of each class and the average minimum distance to each class [15,44].

**Table 1. DHS indicators of interest and their link to malaria.**

| Indicator | Short name | Link to malaria |
|---|---|---|
| **Socioeconomic** | | |
| **Proportion of Fula** | Fula ethnicity | Fula are known to be less susceptible to malaria due to higher antibody titres [32,40]. Cultural beliefs may also influence malaria prevention behaviour [32]. |
| **Stunting in children** | Stunting | Stunting increases the risk of malaria by reducing immunity, although the reverse relationship has sometimes been observed [29,33]. |
| **Anemia prevalence in children** | Anemia | Malaria is a major cause of anemia [41]. |
| **Access to basic sanitation service** | Sanitation | Poor sanitation increases exposure to water-borne diseases (e.g., helminths), which are often co-infected with malaria [34]. |
| **Wealth index** | Wealth index | Household wealth influences access to health and prevention resources, type of housing, education, etc. [42]. |
| **Literacy rate in women** | Literacy | Educated mothers are better informed about malaria prevention and can increase household income through better jobs [29,31]. |
| **Malaria prevention** | | |
| **Proportion of households with at least one ITN** | ITN ownership | Influences the capacity to anticipate malaria by protecting people against mosquito bites. |
| **Proportion of households with at least one ITN for every two people that slept at the house the night before the survey** | ITN ownership for 2 | Influences the capacity to anticipate malaria by protecting people against mosquito bites. |
| **Proportion of population with access to an ITN in their household** | ITN access | Influences the capacity to anticipate malaria by protecting people against mosquito bites. |
| **IRS coverage** | IRS | Influences the capacity to anticipate malaria by killing mosquitoes that land on the sprayed surface. |

*Note.* ITN and IRS stand for insecticide-treated nets and indoor residual spraying, respectively. All indicators are expressed as proportions, except for the wealth index, which is a standardised factor score (mean = 0 and standard deviation = 1) averaged by DHS cluster.

## Modelling approaches

Three categories of methods are tested for modelling DHS indicators: (1) spatial interpolation methods (i.e., inverse distance weighting, thin plate splines, kriging), (2) ensemble methods (i.e., random forest regression) and (3) Bayesian geostatistical models (see Table 3).

   **Spatial interpolation methods.** *Inverse distance weighting*: Inverse distance weighting (IDW) is an exact spatial interpolation method, meaning that the interpolated surface passes exactly through the sample points. It estimates the value ($\hat{z}_0$) of a random variable $z$ at an unsampled location $s_0$ as the weighted average of its values ($z_i$) at the $k$-nearest observations at location $s_i$ ($i = 1, \ldots, k$) [60]:

$$\hat{z}_0 = \frac{\sum_{i=1}^{k} w_i(s_0)\ z_i}{\sum_{i=1}^{k} w_i(s_0)}$$

(1)

where $w_i(s_0)$ is the interpolation weight assigned to each point at location $s_i$ for calculating the interpolated value at location $s_0$ [60]. It is computed as follows:

$$w_i(s_0) = \left|\left| s_0 s_i \right|\right|^{-p}$$

(2)

**Table 2. Geospatial covariates and data sources.**

| Category | Covariates | Data source |
|---|---|---|
| Climate & environment | Precipitation, temperature at 2 m, potential evapotranspiration, near-surface relative humidity, climate moisture index | CHELSA [45] |
| | Day land surface temperature (LST), night LST, daily LST range | MODIS [46,47] |
| | NDVI, NDWI, NDMI | Computed from Sentinel-2 L1C composites of the Joint Research Centre (JRC) [48] |
| Land use & land cover | Proportion and distance to water, trees, flooded vegetation, crops, grass, bare ground, shrubland | Dynamic World by Google and the World Resources Institute [49] |
| | Proportion and distance to settlements | World Settlement Footprint from the German Aerospace Center [50] |
| | Residential built-up surface, building height | Global Human Settlement Layer from the JRC [51,52] |
| | Average and median nighttime lights | Annual VIIRS nighttime lights v2.1 from the Earth Observation Group [53] |
| Remoteness | Distance to major roads, waterways, education facilities and health facilities | OpenStreetMap |
| | Travel time to major cities, walking-only and motorized travel time to healthcare facilities | Malaria Atlas Project [54,55] |
| Demographic | Population counts, births, pregnancies | WorldPop project [56,57] |
| | Proportion of Wolof, Fula, Serer, Diola, Mandingue, Soninke, not Senegalese and other ethnic groups | ETH Zurich [20] |
| Topography | Elevation | US Geological Survey [58] |
| Livestock densities | Density of goat, cattle, pig, poultry, sheep | Gridded Livestock of the World v2.01 by FAO in collaboration with ILRI, the University of Oxford and the Université Libre de Bruxelles [59] |

*Note.* Abbreviations: LST (land surface temperature), NDVI (Normalized Difference Vegetation Index), NDMI (Normalized Difference Moisture Index), NDWI (Normalized Difference Water Index), JRC (Joint Research Centre), VIIRS (Visible Infrared Imaging Radiometer Suite).

**Table 3. Overview of modelling approaches investigated in this study.**

| Covariates | Spatial interpolation methods | Ensemble methods | Bayesian geostatistical models |
|---|---|---|---|
| **No** | Inverse distance weighting<br>Thin plate spline<br>Ordinary kriging | x | x |
| **Yes** | Universal kriging | Random forest regression | Bayesian regression models |

*Note.* The 'Covariates' column indicates whether the modelling approaches use covariates or not. Boxes marked with a cross (x) indicate that we did not use approaches from this category.

where $\lVert s_0 s_i \rVert$ is the Euclidean distance between $s_0$ and $s_i$, and $p$ is the power assigned to that distance. The higher the value of $p$, the less the influence distant observation points have on the target point [60]. The best fit for the number of neighbours ($k$) and the power parameter ($p$) was found using a 50-repeated 4-fold random cross-validation.

**Thin plate spline**: Fitting a thin plate spline (TPS) can be thought of as fitting a thin steel plate through the sample points, and this fit can be more or less smooth; the fit can pass through the sample points exactly, or it can be more flexible and deviate slightly from them [61]. This consists of fitting a function $f$ through the sample points such that the energy required to bend the steel plate is minimised [61]. This is achieved by minimising the following objective function:

$$\sum_{i=1}^{n} (z_i - f(x_i, y_i))^2 + \lambda J[f]$$

(3)

where $x_i, y_i$ are the coordinates of location $s_i$ $(i = 1, \ldots, n)$ in the 2D space, $z_i$ is the value of observation at location $s_i$, $J[f]$ is the penalty function, $\lambda$ is the penalty parameter and $f$ is the function fitted to the observation points. The first term in (3) represents the goodness-of-fit (measured by the squared residuals) and the second term represents the average curvature of the TPS allowed by the penalty parameter $\lambda$ [20]. Function $f$ fitted through the sample points is given by [61]:

$$f(x, y) = a_0 + a_1 x + a_2 y + \sum_{i=1}^{n} c_i \, r_i^2 \ln(r_i)$$

(4)

where $r_i$ is the Euclidean distance between observation at location $s_i$ and the point with coordinates $x, y$, and $a_0, a_1, a_2, c_i$ are unknown coefficients to be estimated. While $a_0, a_1, a_2$ represent the global linear trend of the spline, the TPS radial basis function $r_i^2 \ln(r_i)$ controls the amount of local distortion [20,61]. Solving (3), we can find a closed-form solution for the parameters $a$ and $c$ for a given value of $\lambda$. At any (unsampled) $x, y$ location, the value of the TPS is given by (4) [61]. The penalty parameter was tuned using a 50-repeated 4-fold random cross-validation, as the recommended generalized cross-validation method can lead to unreliably small parameter estimates [62].

*Kriging*: Kriging has also been used in previous work to map DHS indicators [63–65]. It estimates the value of a random variable at unsampled locations as the weighted average of its values at sample points, such as IDW. The difference with IDW is that the weights are estimated based on a variogram function that recovers the spatial autocorrelation pattern in the data [60]. The variogram function $\gamma(h)$ describes how the dissimilarity (semi-variance) between the values of the random variable $z$ at pairs of sample points increases with the distance $h$ separating these points:

$$\gamma(h) = \frac{1}{2}(z(s+h) - z(s))^2$$

(5)

Kriging is a geostatistical method, hence the response variable $z$ is decomposed into a general non-random spatial trend modelled by the expectation $E(z)$ and a spatially correlated random term $R$ that represents the local deviation from this trend:

$$z = E(z) + R$$

(6)

Ordinary and universal kriging differ in the assumptions made about the spatial trend. Ordinary kriging (OK) assumes a stationary and unknown spatial trend. The variogram function hence models both the spatial trend and the residual term [60]. In universal kriging (UK), the spatial trend is not constant over space, but depends on covariates. The global trend is then modelled as a linear combination of covariates:

$$E(z) = X(s_i)\beta$$

(7)

where $X(s_i)$ is the vector of covariates at location $s_i$ $(i = 1, \ldots, n)$ and $\beta$ are the regression coefficients. In this approach, the variogram function is applied only to the spatially correlated residuals, which are then interpolated at unsampled locations using kriging weights. The final interpolated value is the sum of the interpolated residual with the global trend calculated from the covariates [66].

The best set of covariates was selected by fitting separate linear regression models for each DHS indicator and implementing stepwise feature selection. This process identified the combination of features that minimized the AIC (Akaike Information Criterion). To avoid multicollinearity issues, only covariates with a variance inflation factor (VIF) below 5 were retained [15]. Note that ethnicity-related variables were not used to model Fula ethnicity (to avoid circularity). Details on the calculation of interpolation weights and hyperparameter tuning are given in S2 Text.

**Random forest regression.** Random forest (RF) regression is a popular machine learning algorithm that uses ensemble learning. A RF model consists of multiple decision trees that model the relationship between the response variable and a set of covariates [67]. Decision trees are built using bagging, where each tree is trained on a random sample drawn with replacement from the observation dataset. This reduces the correlation between trees and thus prevents overfitting [67]. Originally used for ecological niche modelling, this method has been widely used to model various other spatial phenomena due to several advantages: RF handles large datasets, it is robust to multicollinearity and it handles non-linear relationships.

RF models were built using a 5-repeated 5-fold random cross-validation (i.e., 25 RF models in total). Hyperparameters tuning was performed by further dividing each fold into four sub-folds and fitting 50 RF models to tune (1) the number of covariates used at each node split, (2) the minimum number of observations per terminal node, (3) the fraction of observations used per decision tree and (4) the number of trees in the model [10].

To further address multicollinearity, covariates were selected using a recursive feature elimination. This method iteratively removes the least important covariate from the model until the predictive performance (measured as a root mean square error on the cross-validation test sets) is the highest [68,69]. Covariate importance is measured as the standardised increase in the average Out of Bag (OOB) error after random permutation of the covariate values [68]. A larger increase in OOB error indicates a more important covariate. The latitude and longitude of the DHS cluster locations were added as supplementary covariates to the original set of covariates, as RF models are not explicitly spatial. Note that ethnicity-related variables were not used to model Fula ethnicity (to avoid circularity).

**Bayesian geostatistical models.** Bayesian geostatistical models treat model parameters (e.g., regression coefficients) as random variables with a density distribution [70]. Similarly, each response value in the data is considered sampled from a distribution. By doing so, this approach allows to quantify uncertainties in the Bayesian predictions by constructing the posterior predictive distributions of the response variable [70]. Bayesian geostatistical models have been increasingly used to model health and demographic outcomes in recent years [4,11,12,15,18].

Estimating the posterior distributions of model parameters can be very computationally intensive for large spatial datasets. The integrated nested Laplace approximation (INLA) approach for latent Gaussian models now allows us to approximate the posterior distributions, significantly reducing the computational time compared to classical Markov chain Monte Carlo (MCMC) approaches [71]. In spatio-temporal models, INLA Bayesian inference is combined with the stochastic partial differential equation (SPDE) approach [72].

In this paper, Bayesian generalised linear models were implemented using INLA-SPDE within the *R-INLA* package [71], with the DHS indicators as response variables. The models can be summarised as follows:

$$z(s_i) = X(s_i)^T \beta + u(s_i) + \varepsilon(s_i)$$

$$u(s_i) \sim GP(0, \Sigma) \tag{8}$$

where $z(s_i)$ is the realisation of a spatial field at cluster location $s_i$ ($i = 1, \ldots, n$), $X(s_i)$ is the vector of covariates at location $s_i$ and $\beta$ are the regression coefficients. $u(s_i)$ is a vector of SPDE random effects representing the structured spatial random variation in $z(s_i)$. It is modelled as a zero-mean Gaussian Process (GP) with Matérn covariance matrix $\Sigma$ [71]. Spatial dependence between two observations of the GP at locations $s_1$ and $s_2$ is modelled using the Matérn covariance function, defined as follows:

$$C(s_1, s_2) = \frac{2^{1-\nu} \sigma_u^2}{\Gamma(\nu)} (\kappa h)^\nu K_\nu(\kappa h) \tag{9}$$

where $h$ is the Euclidean distance between $s_1$ and $s_2$, $\sigma_u^2$ is the marginal variance, $\nu > 0$ controls the smoothness of the spatial process, $K_\nu$ is the modified Bessel function of the second kind and $\kappa > 0$ is a scale parameter controlling the range,

i.e., the distance from which there is no more spatial autocorrelation in the data. The unexplained random variation is included in the term $\varepsilon(s_i) \sim N(0, \sigma_e^2)$.

All DHS indicators representing proportions (Table 1) were modelled using a binomial likelihood, with a zero-inflated binomial likelihood for IRS given the high prevalence of zero values. A Gaussian likelihood was used for the wealth index. As with UK, covariate selection was performed using stepwise selection on linear regression models (excluding ethnicity-related variables for modelling Fula ethnicity). Multicollinearity was addressed by checking the VIF. Details on the SPDE approach and prior specification are provided in S2 Text.

## Model validation and predictions

Model validation followed the validation scheme of [3], which is also used to compare different methods. DHS data were divided into a training and a test set using an 80–20 split ratio. All methods were trained on the training set, with cross-validation applied for hyperparameter tuning when necessary (as detailed in each method section). Predictive performance was evaluated on the 20% test set using (i) the root mean square error (RMSE), (ii) the mean absolute error (MAE), and (iii) the $R^2$ (coefficient of determination) calculated as follows [3]:

$$RMSE = \sqrt{\frac{\sum_{i=1}^{n}(z_i - \hat{z}_i)^2}{n}}$$

$$MAE = \frac{\sum_{i=1}^{n}|z_i - \hat{z}_i|}{n}$$

$$R^2 = 1 - \frac{\sum_{i=1}^{n}(z_i - \hat{z}_i)^2}{\sum_{i=1}^{n}(z_i - \bar{z})^2}$$

(10)

where $n$ is the number of observations, $z_i$ is the value of observation $i$, $\hat{z}_i$ is its predicted value and $\bar{z}$ is the mean value over all observations. The RMSE and MAE are restricted to positive values, with lower values indicating better performance. The $R^2$ ranges from minus infinity to 1, with higher values indicating better performance. For Bayesian geostatistical models, metrics were calculated using the posterior mean estimates. Predictive maps of DHS indicators (Table 1) were generated by each method on a common 1x1 km resolution grid.

Finally, all methods were compared using the following criteria: predictive performance (RMSE, MAE and $R^2$ [3]), restriction of the prediction range to the observation range, sensitivity to the number and spatial distribution of observations, uncertainty quantification, computational efficiency, need for covariate collection and processing, ease of implementation/intuitiveness and handling of missing response data.

## Results

### Model validation

Model predictive performance was evaluated on the test set and measured with the RMSE, MAE and $R^2$ [3]. $R^2$ values are summarised in Table 4, Figs 1 and 2. RMSE and MAE scores are given in S1 Table. Some DHS indicators could be accurately predicted, with best $R^2$ values between 0.62 and 0.86 (literacy, sanitation, Fula ethnicity, IRS and wealth index). Only two indicators were not accurately predicted, with $R^2$ values around 0.30 (ITN ownership for 2 and anemia). For the remaining indicators, the models achieved medium predictive performance, with best $R^2$ scores ranging from 0.40 to 0.49 (ITN access, stunting and ITN ownership) (Table 4). The RMSE and MAE scores also indicate overall good model performance, with RMSE values ranging from 0.087 to 0.194 for all proportion-type indicators and 0.330 for the wealth index. MAE values ranged from 0.044 to 0.150 for all proportion-type indicators and 0.258 for the wealth index (S1 Table).

**Table 4. R² values for all modelling approaches and all DHS indicators.**

| Indicator | Covariate-independent | | | Covariate-based | | | Best diff. |
|---|---|---|---|---|---|---|---|
| **Socioeconomic** | **IDW** | **TPS** | **OK** | **UK** | **RF** | **BM** | |
| **Fula ethnicity** | 0.61 | **0.66** | **0.66** | 0.62 | 0.60 | 0.53 | - 0.04 |
| **Stunting** | 0.35 | 0.32 | 0.34 | 0.30 | **0.42** | 0.20 | + 0.07 |
| **Anemia** | 0.23 | 0.25 | 0.25 | 0.29 | **0.30** | 0.26 | + 0.05 |
| **Sanitation** | 0.37 | 0.41 | 0.40 | 0.51 | **0.63** | 0.58 | + 0.22 |
| **Wealth index** | 0.44 | 0.49 | 0.58 | 0.64 | **0.86** | 0.68 | + 0.28 |
| **Literacy** | 0.40 | 0.35 | 0.43 | 0.61 | 0.56 | **0.62** | + 0.19 |
| **Malaria prevention** | **IDW** | **TPS** | **OK** | **UK** | **RF** | **BM** | **Best diff.** |
| **ITN ownership** | 0.39 | **0.49** | 0.31 | 0.45 | 0.42 | 0.48 | - 0.01 |
| **ITN ownership for 2** | 0.22 | 0.27 | 0.24 | 0.26 | 0.18 | **0.29** | + 0.02 |
| **ITN access** | 0.36 | **0.40** | 0.29 | 0.31 | 0.31 | 0.26 | - 0.09 |
| **IRS** | 0.65 | 0.67 | 0.70 | **0.71** | 0.56 | **0.71** | + 0.01 |

*Note.* R² values are evaluated on the test set (20% of the data). Higher R² values indicate better model performance. The best R² values are marked in bold and underlined. The last column 'Best diff.' shows the difference between the best covariate-based model and the best covariate-independent model. Positive values indicate that the covariate-based model performed best, while negative values indicate that the other model was the most performant. Abbreviations: IDW (inverse distance weighting), TPS (thin plate spline), OK (ordinary kriging), UK (universal kriging), RF (random forest), BM (Bayesian model), Best diff. (best difference), ITN (insecticide-treated net), IRS (indoor residual spraying).

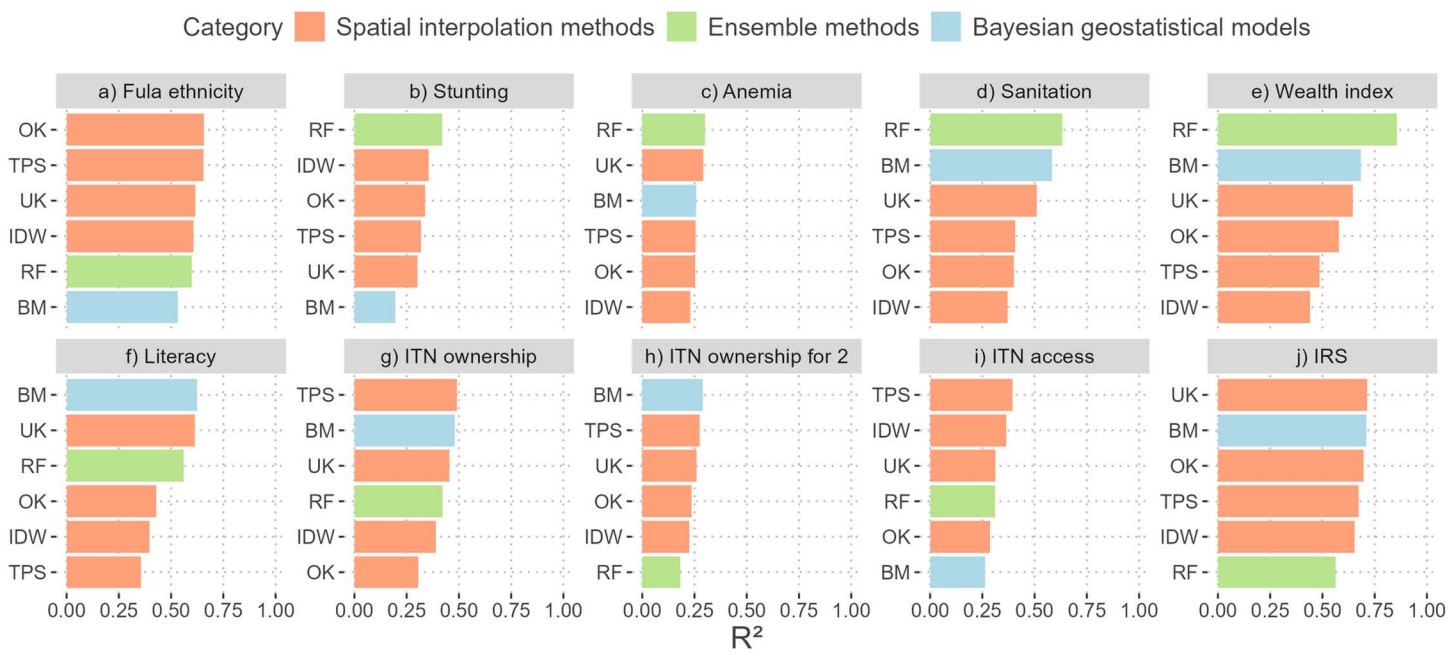

**Fig 1. Modelling approaches ranked by R² values.** R² values are evaluated on the test set (20% of the data). R² is the coefficient of determination (values range from minus infinity to 1), with higher values indicating better model performance. Orange, green and blue colours respectively indicate spatial interpolation methods, ensemble methods and Bayesian geostatistical models. Abbreviations: IDW (inverse distance weighting), TPS (thin plate spline), OK (ordinary kriging), UK (universal kriging), RF (random forest), BM (Bayesian model), ITN (insecticide-treated net), IRS (indoor residual spraying).

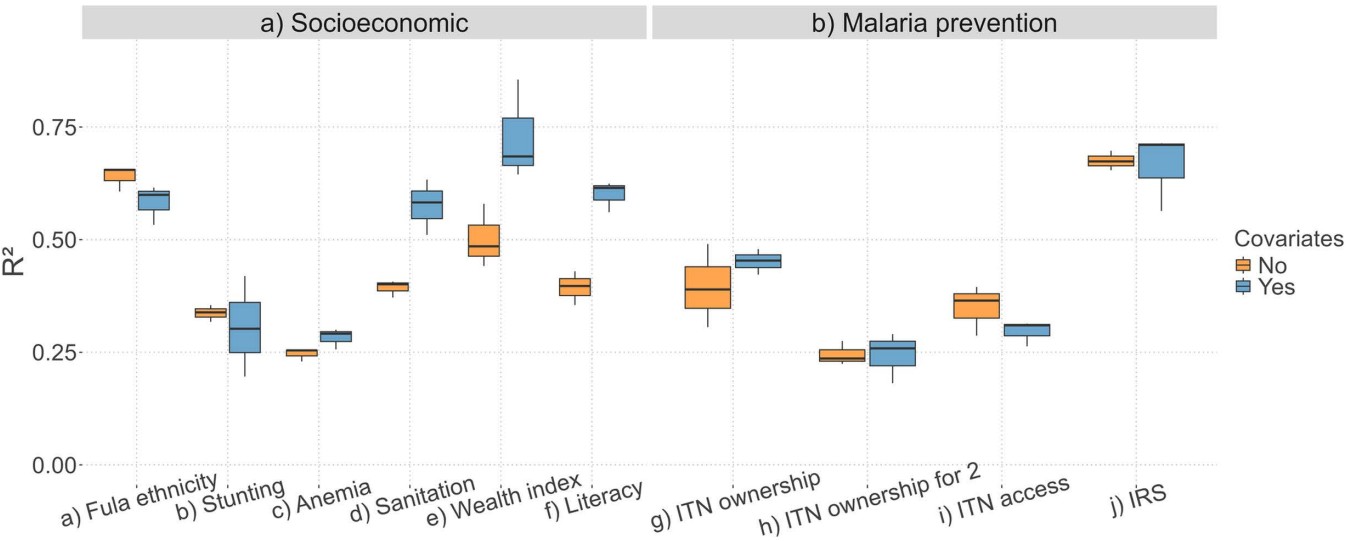

**Fig 2. Boxplots of R² values.** DHS indicators are grouped by socioeconomic (a) and malaria prevention (b) categories. For each indicator, the R² values from all methods are combined to create the boxplot. R² values are evaluated on the test set (20% of the data). R² is the coefficient of determination (values range from minus infinity to 1), with higher values indicating better model performance. Blue and orange colours respectively indicate covariate-based models and covariate-independent models. Abbreviations: ITN (insecticide-treated net), IRS (indoor residual spraying).

None of the three categories of methods emerged as the best performer (Fig 1 and S1 Table). For all socioeconomic indicators except Fula ethnicity, the best performing method always used covariates and was most often RF (Table 4 and S1 Table). In addition, RF was always in the top 3 methods for socioeconomic indicators (Fig 1b–1f and S1 Table), except for Fula ethnicity, for which TPS and OK performed best. Compared to socioeconomic indicators, malaria prevention indicators showed more variation in the best performing methods, which ranged between TPS (based on spatial autocorrelation), BM and UK (both using covariates) (Table 4). When BM and UK performed best, they only slightly outperformed methods without covariates (TPS or OK), with R² differences of up to 0.02 (see column 'Best diff.' in Table 4). In comparison, for socioeconomic variables, covariate-based methods outperformed covariate-independent methods by up to 0.28 in R². These results suggest that methods that rely only on spatial autocorrelation tend to perform better in modelling malaria prevention indicators than socioeconomic indicators (with the exception of Fula ethnicity). Fig 2 further highlights these findings, with covariate-based methods showing higher R² values for socioeconomic variables (Fig 2a). For malaria prevention indicators, there is less difference between the performance of covariate-independent and covariate-based methods (Fig 2b). Note that these trends are also consistent with the RMSE and MAE values (S1 Table).

## Geospatial covariates

The results of covariate selection for UK, RF and BM are detailed in S2 Table. On average, RF used 28 covariates, whereas BM and UK used only 12 (S2 Table). These methods selected different types of covariates. Some covariates were selected for most DHS indicators by RF (e.g., climatic variables, longitude, nighttime lights), while UK and BM used more land cover variables (e.g., proportion of crops). A few variables were used in all models, in particular accessibility variables such as walking time to health facilities and distance to major roads (S2 Table), which are key variables influencing health indicators. It is expected that different methods would select different variables; while RF handles multicollinearity [67], UK and BM require multicollinearity checks prior to modelling. RF deals with non-linear relationships between the

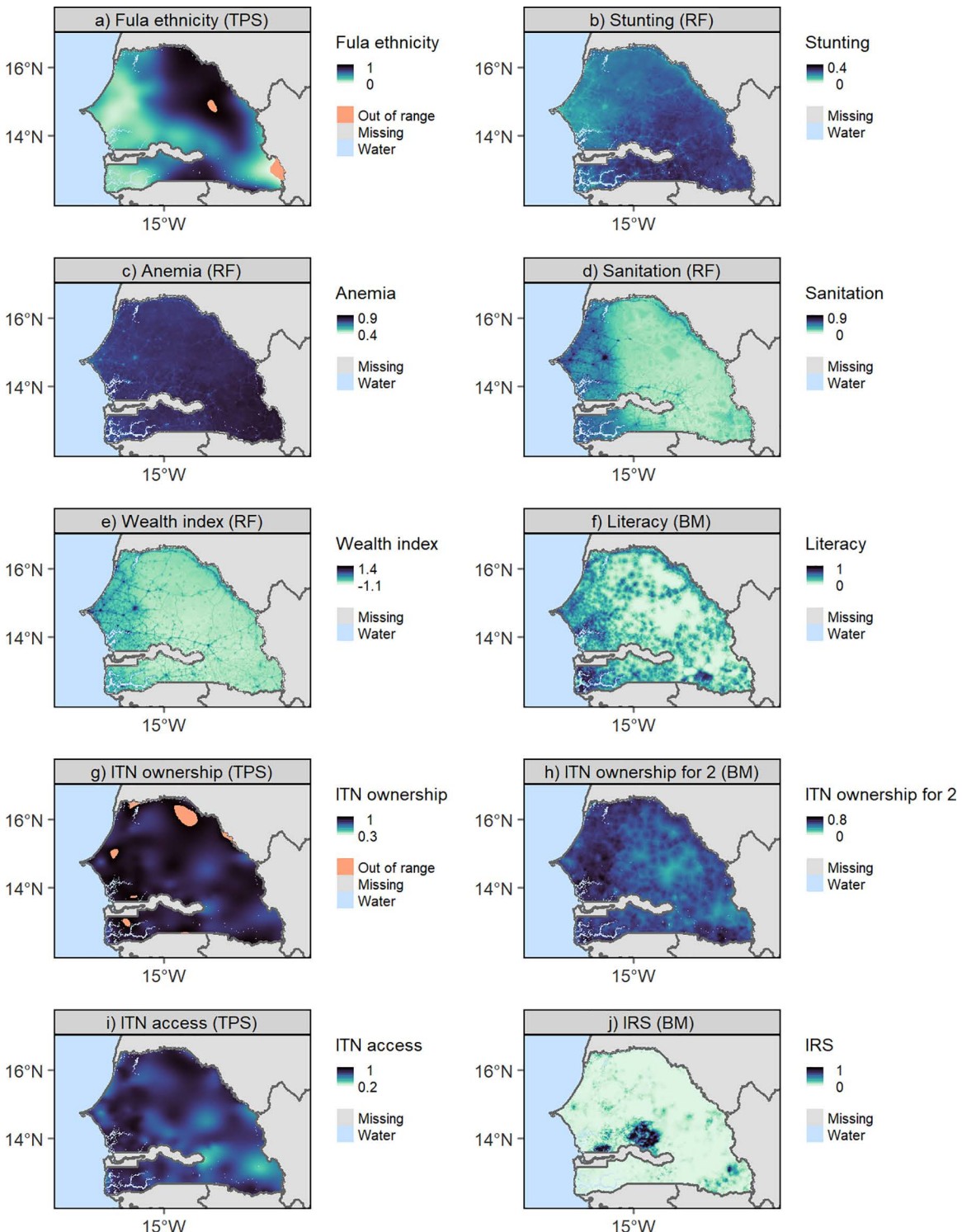

**Fig 3. Predicted DHS indicators at 1x1 km for Senegal.** For each indicator, the best performing method was used, that is, thin plate splines (Fula ethnicity, ITN ownership, ITN access), random forest (stunting, anemia, sanitation, wealth index) and Bayesian models (literacy, ITN ownership for 2, IRS). 'Out of range' indicates predicted values that are outside the possible range of values of the indicator (below 0 or above 1 for proportions). National boundaries were downloaded from GADM. Abbreviations: TPS (thin plate spline), RF (random forest), BM (Bayesian model), ITN (insecticide-treated net), IRS (indoor residual spraying).

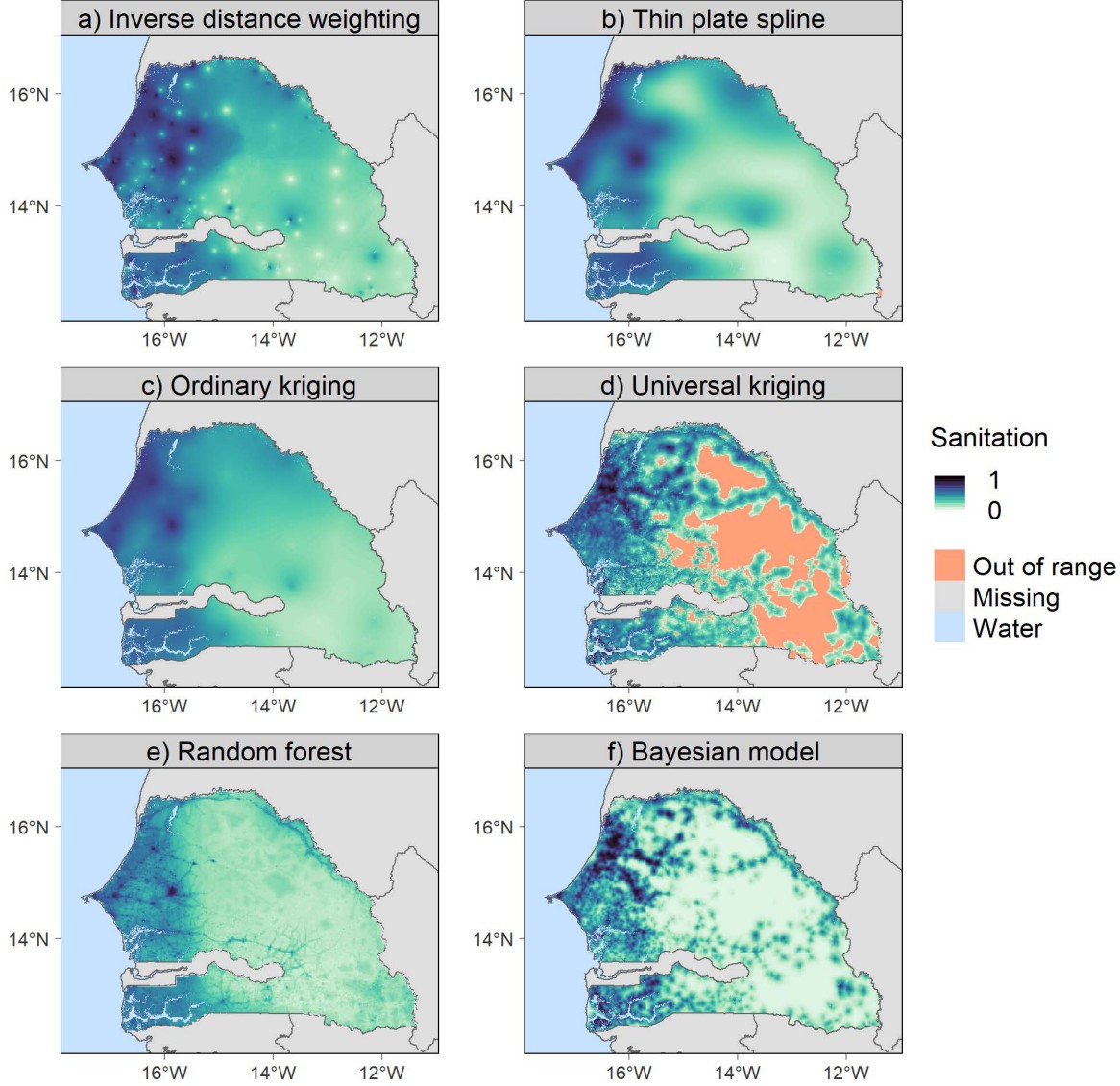

**Fig 4. Predicted access to basic sanitation service at 1x1 km for Senegal.** The maps show the spatial distribution of the proportion of households with access to basic sanitation in Senegal. 'Out of range' indicates predicted values that are outside the possible range of values of the indicator (below 0 or above 1). Out-of-range predictions were made by thin plate spline and universal kriging. National boundaries were downloaded from GADM.

response variable and the covariates [67], whereas the feature selection for UK and BM was based on linear regression models. Lastly, unlike UK and BM, RF does not explicitly account for spatial autocorrelation.

## Predictive maps

We produced predictive maps of the DHS indicators at a resolution of 1 km in Senegal. Fig 3 shows maps of all DHS indicators predicted using their respective best performing method. Most indicators show a west-east gradient, with better socioeconomic status and access to malaria prevention in the western regions. Fula ethnicity and IRS do not follow this pattern and are instead highly clustered in space.

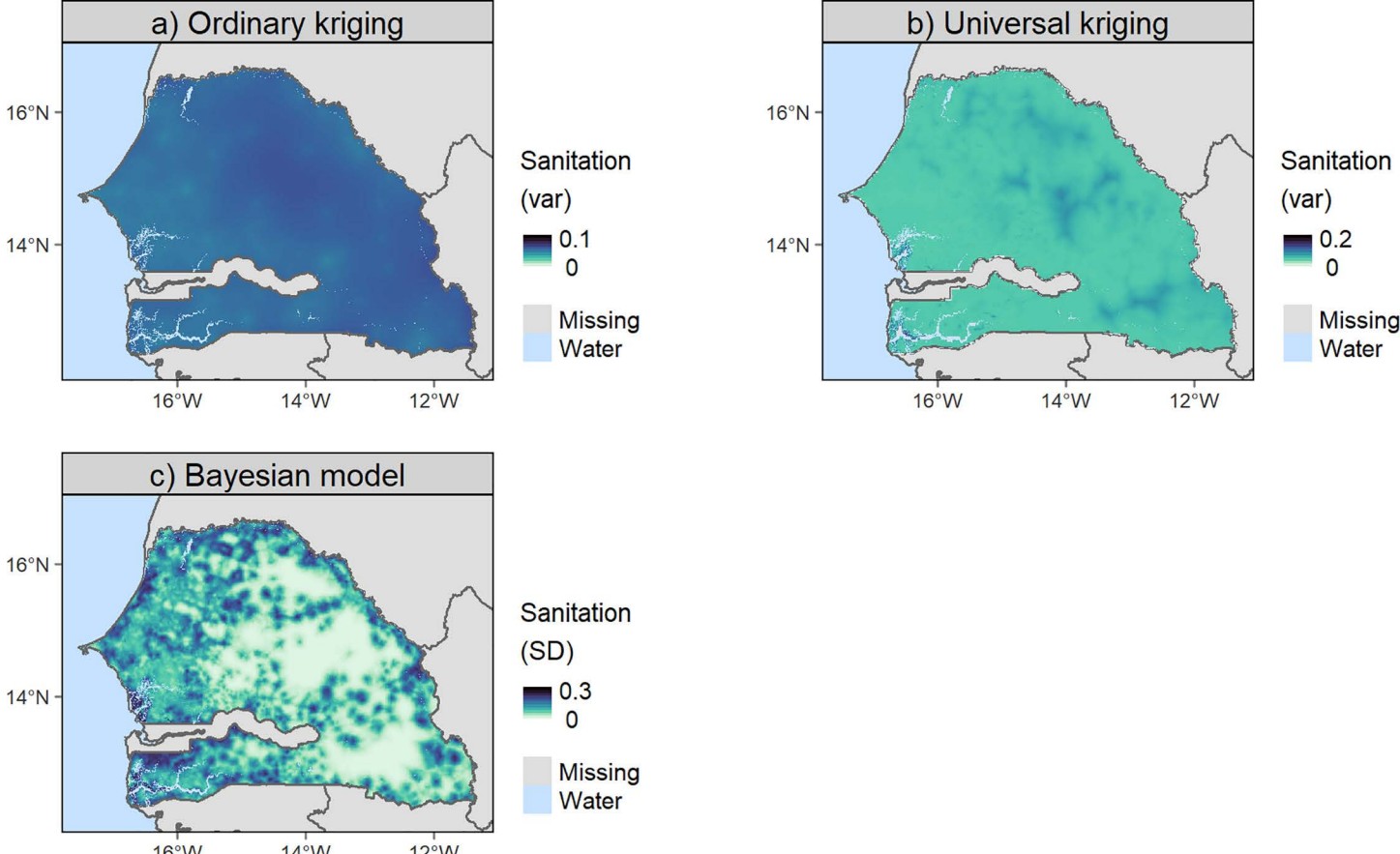

**Fig 5. Uncertainty maps of the predicted access to basic sanitation service.** Uncertainty is measured as a prediction variance (var) for kriging methods (a, b) and as a standard deviation (SD) for Bayesian models (c). Higher values of SD or variance indicate areas with greater uncertainty in the predicted access to basic sanitation, reflecting lower confidence in the accuracy of the predictions. National boundaries were downloaded from GADM.

To compare the predictions between methods, each indicator was also predicted using all the methods in this study. Fig 4 shows, as an example, the predicted access to basic sanitation service at 1x1 km for Senegal. Similar maps can be found for other DHS indicators in the electronic supplementary material (S1–S18 Figs). All maps in Fig 4 show a similar pattern of predicted access to basic sanitation service, with a decreasing gradient from west to east. However, due to the specificities of each method and their relative performance (see Table 4), the maps show differences in predictions at a finer scale. RF and BM allowed a higher level of detail (Fig 4e and 4f) due to the use of covariates compared to TPS and OK (Fig 4b and 4c), which predicted smoother interpolation surfaces. TPS and UK predicted out-of-range values, i.e., values outside the possible range for the indicator (from 0 to 1 as it is a proportion) (Fig 4b and 4d). Kriging and Bayesian models allowed to quantify the uncertainty in the predictions as the prediction variance and standard deviation respectively, see Fig 5 as an example.

### Comparison of methods

Although predictive performance is an essential feature for model comparison, there are other criteria to consider. These comparison criteria are summarised in Table 5 and discussed in the following sections.

**Table 5. Criteria for comparing modelling approaches.**

| Criteria | IDW | TPS | OK | UK | RF | BM |
|---|---|---|---|---|---|---|
| **High predictive performance** | – | ± | ± | ± | + | + |
| **Predictions always within range of observations** | + | – | – | – | + | + |
| **Low sensitivity to the number and spatial distribution of the observations** | – | – | ± | ± | + | + |
| **Quantification of uncertainty** | – | – | + | + | – | + |
| **High computational efficiency** | + | + | ± | ± | – | – |
| **No covariate collection & processing** | + | + | + | – | – | – |
| **High ease of implementation/intuitiveness** | + | + | ± | ± | – | – |
| **Effective handling of missing response data** | – | – | – | – | ± | + |

*Note.* The symbols illustrate how the methods perform against each criterion: + indicates strong performance, ± indicates moderate performance, - indicates poor performance. Abbreviations: IDW (inverse distance weighting), TPS (thin plate spline), OK (ordinary kriging), UK (universal kriging), RF (random forest), BM (Bayesian model).

IDW is straightforward, intuitive and computationally efficient. Besides, the interpolated values are constrained to the range of observed values because the interpolation weights always lie between 0 and 1 (and sum to 1) [60]. However, as weights relies solely on distance and do not consider the spatial pattern in the data, IDW is highly sensitive to the spatial distribution and number of observations. In areas with dense observations, the sample points used for interpolation may be close to the target point. In areas with sparse observations, neighbouring sample points may be located at large distances, leading to inaccurate predictions [60]. IDW does not quantify prediction uncertainty and does not account for missing data in the response variable.

TPS can be implemented easily, without any covariate processing. Besides, it allows to recover both completely linear and highly convex surfaces [20]. A drawback of this flexibility is that TPS may overfit noise in the data, leading to less reliable predictions in certain cases. If the spatial distribution of sample points is irregular, TPS may overfit in regions with dense data points and oversmooth in regions with sparse data. In addition, TPS can predict values largely outside the plausible range of values, i.e., the so-called Runge effect, where the interpolated surface oscillates between the sample points and 'overshoots' [60]. TPS made out-of-range predictions for four indicators (Fig 4b and S9b, S15b and S17b Figs).

Kriging offers an improvement over IDW for estimating interpolation weights by using the spatial autocorrelation pattern in the response variable. The prediction variance helps to assess the uncertainty of the predictions. High prediction variance means that the predicted value is based on sample points located at greater distances. Kriging is more robust to the relative distribution and number of observations [60], provided that there are enough to recover the spatial pattern. They sometimes predict out-of-range values due to negative weights when the dataset contains both highly clustered and more distant observations [73]. Local kriging methods could compensate for negative weights, but are computationally inefficient [20]. OK and UK made out-of-range predictions for one (S15c Fig) and seven indicators respectively (Fig 4d and S3d, S7d, S9d, S11d, S15d and S17d Figs).

RF deals with multicollinearity and non-linear relationships between covariates and the response variable [67]. Based on decision trees, they implicitly capture interactions between covariates. As non-spatial models, they are less sensitive to the spatial distribution of observations than previous methods, provided there are enough observations to recover the relationships between the response and the covariates. In addition, predictions are constrained to the range of observations. However, RF are computationally intensive when tuning the hyperparameters and performing stepwise covariate selection. They are less intuitive than IDW, TPS and kriging, and do not quantify prediction uncertainty. The standard RF model does not handle missing values in the response, but some recent implementations (e.g. missForest) allow imputation of missing data at model fit [74].

By defining priors and a likelihood in the model, Bayesian models allow predictions to be constrained to a range of values, avoiding out-of-range predictions [70]. If the model is robust, the estimation of the posterior distributions of the parameters is less affected by the removal or addition of observations in the model. In addition, methods that use covariates are less sensitive to uneven spatial distribution of sample points, because even if they fail to recover the spatial autocorrelation pattern, they also depend on the covariates to model the response variable. Uncertainty can be quantified through standard deviations, 2.5th and 97.5th percentiles of the posterior distribution. Unlike previous methods, Bayesian models handle missing response data and provide imputation during model fitting. However, they can be time consuming with lots of covariates and may be less intuitive than other models when lacking statistical skills.

## Discussion

Achieving the SDG 2030 agenda requires fine-resolution mapping of SDG indicators to monitor progress and support targeted health interventions [3,4]. In this context, DHS data have been widely used to map SDG indicators across countries and over time, using a variety of methods in terms of complexity and model inputs [4,8,11–13,15,17–20]. In this paper, we compared three categories of methods to model DHS indicators: (1) spatial interpolation methods (i.e., IDW, TPS, OK, UK), (2) ensemble methods (i.e., RF), and (3) Bayesian geostatistical models (BM). Focusing on SDG Target 3.3, we applied these methods to map socioeconomic and malaria prevention indicators at 1 km resolution in Senegal.

Our findings show that most indicators could be mapped with medium to high predictive accuracy, with $R^2$ values ranging from 0.40 to 0.86. This is in line with previous studies modelling DHS indicators in African countries [3,9,15]. However, there was no consensus on the best method or category of methods. RF was the best method for modelling most socioeconomic indicators. Previous studies have already highlighted its good performance with DHS data (e.g., malaria prevalence [75], wealth index [8,19]). For malaria prevention indicators, there was more diversity in the best performing approaches (ranging between TPS, UK and BM). Increasing model complexity did not always improve predictive performance, e.g., TPS sometimes outperformed RF or Bayesian geostatistical models (e.g., ITN access in Table 4), as in previous work [25]. Previous research that estimated spatial accessibility to health facilities has questioned the use of data-demanding methods when (outcome) data are of poor quality [76]. This may be the case here, where the displacement of DHS cluster coordinates for anonymisation affects the spatial resolution of the data [38]. Nevertheless, other research has shown that Bayesian models can accurately model DHS indicators in various settings [3,4,12,15,18,24]. Overall, our conclusion that no single method consistently excels in all contexts aligns with [3], which found no best approach between Bayesian geostatistical models and artificial neural networks. Similarly, [25] highlights that model performance depends on factors such as the study area, data scarcity and the structure of the spatial pattern.

Although there was no clear best method, we found that socioeconomic indicators were generally better predicted using covariates, while methods relying only on spatial autocorrelation performed better for Fula ethnicity and malaria prevention indicators. This may be explained by the spatial clustering of both ethnicity and access to malaria prevention, which may be better captured by methods recovering spatial autocorrelation patterns. Ethnic groups may be spatially clustered due to historical factors such as migration patterns and settlement practices (see the distribution of Fula ethnicity in Fig 3a). Access to malaria prevention may be influenced by policy decisions on intervention planning, resulting in strong spatial autocorrelation in areas (e.g., administrative unit, health district) where interventions are implemented. This may explain why TPS performed well for these indicators. The interpolated surface can adapt to both abrupt changes, capturing local variations (by passing through the sample points), and smooth trends (by deviating from the sample points) [20,61]. This makes TPS particularly suitable for datasets with heterogeneous spatial structures (e.g., variable values change abruptly in space, as seen with intervention-related variables and ethnic groups). Another explanation for the lower performance of covariate-based methods could be because not all health variables are environmentally linked [9]. While socioeconomic status is more stable over time, access to malaria prevention can change rapidly following interventions such as ITN distribution, making it less predictable by long-term covariates. The lack of association between malaria

prevention access and covariates may also be due to seasonal effects, as prevention behaviour and access peak during the rainy season, which was not accounted for in the covariates. The high cloud cover in Senegal during the rainy season limits the availability of satellite imagery [48], making it challenging to compile covariates from the wet season. Annual composites (e.g., NDVI and NDWI) may therefore better represent the dry season [48], missing temporary water bodies and other seasonal features that correlate with malaria risk.

This paper also provides a list of criteria that can be used to compare models for mapping DHS indicators and health and demographic variables in general. Constraining predicted values to the observed range is essential, as highly accurate maps can be useless with out-of-range predictions. Low sensitivity to the number and spatial distribution of observations is also important; overall, covariate-based methods may be less sensitive because the structure of the response variable is also captured by the covariates. Distance-based approaches such as IDW will typically be less accurate in areas with sparse observations [60]. Computational efficiency, covariate processing and intuitiveness of the methods also need to be considered. Where complex approaches perform better, the trade-off between complexity and information gain might be assessed. For example, RF sometimes only slightly improved predictive performance over IDW (e.g., see ITN ownership in Table 4). However, RF requires more computational resources and storage to handle large covariate datasets, whereas IDW can be implemented in GIS software without coding or extensive covariate processing. Lastly, the ability to quantify the uncertainty in predictions and handle missing data are other interesting features to consider. These criteria were used to compare the methods employed in this study. IDW, for instance, is an intuitive, computationally efficient method that does not predict out-of-range values [60]. However, it is sensitive to the spatial distribution of observations and cannot quantify prediction uncertainty or handle missing data effectively. TPS offers more flexibility, being able to model both linear and highly convex surfaces [20], but it can overfit data and become less reliable, especially with irregular spatial distributions of sample points. In some cases, it can lead to out-of-range predictions, known as the Runge effect. Kriging improves upon IDW by incorporating spatial autocorrelation into the interpolation process and quantifying prediction uncertainty [60]. This makes it more robust to uneven spatial distributions of observations, although it can sometimes lead to out-of-range predictions [73]. RF handles non-linear relationships with covariates and multicollinearity [67], and is less sensitive to the spatial distribution of observations than spatial models. Predictions are constrained by the range of observations, but RF is computationally intensive and does not quantify prediction uncertainty. Bayesian geostatistical models constrain predictions to a defined range, handle missing data inherently and allow uncertainty to be quantified [70]. They are less sensitive to the number of observations and spatial distribution, but are computationally intensive and require statistical expertise.

The choice of method for predicting DHS indicators depends on the purpose of the predictive maps. Planning health interventions may require accurate maps with uncertainty estimates, in which case Bayesian geostatistical models are appropriate. Yet, simpler approaches may sometimes be preferred to complex geostatistical models. Where computational resources and time allow, or where relationships with covariates are of interest, Bayesian or RF models can also be used. Maps used for communication (e.g., reports) or as covariates to model other variables could be based on simpler methods such as IDW or Bayesian models without covariates to avoid circularity issues. Simpler methods can also be used if the main objective is to identify hotspots or coldspots (e.g., areas with low access to malaria prevention) rather than to obtain the actual estimates. Research that aims to model multiple outcomes simultaneously could investigate joint Bayesian models, which account for correlations between the spatial structures of all response variables. However, these models are significantly more complex and require more computational resources. Overall, the results of this study suggest using covariate-based models for socioeconomic indicators and methods based on spatial autocorrelation for variables with heterogeneous spatial structures, such as malaria prevention or ethnicity. We do not recommend using methods that predict out-of-range values (TPS and kriging) as out-of-range predictions are useless for decision-making. Future work will investigate TPS with tension or combine TPS with other interpolation methods (see [20]) to avoid out-of-range predictions. Similarly, kriging could incorporate corrections for negative weights, see the algorithms developed in [73].

This study has some limitations related to the data and methods used. Although the DHS data were collected during the rainy season, the survey covers several months, during which malaria transmission and prevention behaviours may vary. Seasonal indicators, such as malaria prevention, may therefore be less accurately captured by DHS. Besides, IRS coverage was better predicted than ITN-related indicators (Table 4), probably because IRS refers to households sprayed in the year prior to the survey, whereas ITN-related indicators are measured at the time of the survey, which varies spatially as the DHS is conducted in the country. Other limitations associated with DHS are the displacement of survey cluster coordinates and the fact that DHS are designed to be representative at the level of coarse administrative areas, rather than at the cluster level [10]. Future work could address these limitations by incorporating the month of data collection, the survey design and the uncertainty in cluster coordinates [77] into the Bayesian geostatistical models. Bayesian models could also include interactions between covariates and random effects to account for urban/rural character of DHS clusters and potential non-linear relationships with other covariates, although this would increase computational costs. Covariate-based models could also benefit from covariates on the availability and quality of malaria services at health facilities, obtained from the Service Provision Assessment surveys of the DHS program. Note that this study does not raise ethical concerns as the DHS data is provided after anonymisation and the output maps are model outcomes with a spatial resolution of 1km.

Overall, our results show that DHS data can be used to produce interpolated surfaces of vulnerability factors that influence malaria risk. Such interpolated surfaces can support policy makers in planning malaria control interventions by showing areas with poor access to malaria prevention. ITN and IRS coverage maps can then be used to target ITN and IRS interventions in Senegal to help achieve the 2030 elimination target. Furthermore, this work can be extended to other vector-borne diseases, such as dengue, for which epidemiological data are not readily available, making maps of vulnerability indicators even more important for intervention planning. Although we focused only on indicators from the Senegal 2017 DHS, findings of this study may be useful in other settings. The results may be generalisable to other DHS surveys, as these are conducted according to standardised protocols and are comparable across countries [10]. In addition, we found that methods based on spatial autocorrelation performed better for malaria prevention and ethnicity variables due to their heterogeneous spatial structures. This is likely to be true for other variables with similar structures (e.g., vaccination coverage, which also depends on policy decisions, or prevalence of female genital mutilation, which is related to ethno-cultural factors [23]). Lastly, the strengths, weaknesses and recommendations regarding the methods used in this study, along with the criteria for choosing a method and the codes provided with this paper, are relevant for modelling other DHS indicators and demographic and health variables in general. We encourage future research to replicate the methods of this study to other countries and to additional malaria-related indicators, such as access to health care and the prevalence of migrants, to further validate and extend these findings.

## Conclusions

In this paper, we compared three categories of methods for modelling DHS indicators: (1) spatial interpolation methods, (2) ensemble methods, and (3) Bayesian geostatistical models. We focused on DHS indicators that are potential drivers of malaria in Senegal, i.e., socioeconomic and malaria prevention indicators. Our main results show that there was no consensus on the best method or category of methods for modelling all indicators. Overall, socioeconomic indicators were better predicted by covariate-based models, while malaria prevention access and ethnicity variables tended to be better predicted by methods relying (only) on spatial autocorrelation. Beyond the predictive performance, there are other criteria to consider when mapping DHS indicators, such as the ability to constrain the range of predicted values or the intuitiveness of the method, and the criteria to consider when choosing a method depend on the end application of the predictive maps. We encourage future research to replicate the methods of this study using other DHS datasets in Senegal and in other countries where DHS are available.

## Supporting information

**S1 Text. Description of DHS indicators and geospatial covariates.**
(DOCX)

**S2 Text. Details on kriging and Bayesian geostatistical models.**
(DOCX)

**S1 Table. RMSE and MAE values (training and cross-validation) for all modelling approaches and all DHS indicators.**
(DOCX)

**S2 Table. Covariates selected by RF, UK and BM for all DHS indicators.**
(DOCX)

**S3 Table. Model parameter values for all methods.**
(DOCX)

**S1 Fig. Predicted stunting in children at 1x1 km for Senegal with each modelling approach.** The maps show the spatial distribution of the proportion (ranging from 0 to 1) of children under 5 years old that are moderately or severely stunted in Senegal. National boundaries were downloaded from GADM.
(TIF)

**S2 Fig. Uncertainty maps of the predicted stunting in children.** Uncertainty is measured as a prediction variance (var) for kriging methods (a, b) and as a standard deviation (SD) for Bayesian models (c). Higher values of SD or variance indicate areas with greater uncertainty in the predicted indicator, reflecting lower confidence in the accuracy of the predictions in these regions. National boundaries were downloaded from GADM.
(TIF)

**S3 Fig. Predicted anemia prevalence in children at 1x1 km for Senegal with each modelling approach.** The maps show the spatial distribution of the proportion (ranging from 0 to 1) of children with mild, moderate or severe anemia in Senegal. Gridded surfaces are produced at a resolution of 1x1 km for all methods examined in the study. The 'Out of range' label indicates predicted values that are outside the possible range of values of the indicator (below 0 or above 1). Out-of-range predictions were made by universal kriging. National boundaries were downloaded from GADM.
(TIF)

**S4 Fig. Uncertainty maps of the predicted anemia prevalence in children.** Uncertainty is measured as a prediction variance (var) for kriging methods (a, b) and as a standard deviation (SD) for Bayesian models (c). Higher values of SD or variance indicate areas with greater uncertainty in the predicted indicator, reflecting lower confidence in the accuracy of the predictions in these regions. National boundaries were downloaded from GADM.
(TIF)

**S5 Fig. Predicted wealth index at 1x1 km for Senegal with each modelling approach.** The maps show the spatial distribution of the household wealth index in Senegal. Gridded surfaces are produced at a resolution of 1x1 km for all methods examined in the study. National boundaries were downloaded from GADM.
(TIF)

**S6 Fig. Uncertainty maps of the predicted wealth index.** Uncertainty is measured as a prediction variance (var) for kriging methods (a, b) and as a standard deviation (SD) for Bayesian models (c). Higher values of SD or variance indicate

areas with greater uncertainty in the predicted indicator, reflecting lower confidence in the accuracy of the predictions in these regions. National boundaries were downloaded from GADM.
(TIF)

**S7 Fig. Predicted literacy rate in women at 1x1 km for Senegal with each modelling approach.** The maps show the spatial distribution of the proportion (ranging from 0 to 1) of women who are literate in Senegal. Gridded surfaces are produced at a resolution of 1x1 km for all methods examined in the study. The 'Out of range' label indicates predicted values that are outside the possible range of values of the indicator (below 0 or above 1). Out-of-range predictions were made by universal kriging. National boundaries were downloaded from GADM.
(TIF)

**S8 Fig. Uncertainty maps of the predicted literacy rate in women.** Uncertainty is measured as a prediction variance (var) for kriging methods (a, b) and as a standard deviation (SD) for Bayesian models (c). Higher values of SD or variance indicate areas with greater uncertainty in the predicted indicator, reflecting lower confidence in the accuracy of the predictions in these regions. National boundaries were downloaded from GADM.
(TIF)

**S9 Fig. Predicted ITN ownership at 1x1 km for Senegal with each modelling approach.** The maps show the spatial distribution of the proportion (ranging from 0 to 1) of households with at least one insecticide-treated net (ITN) in Senegal. Gridded surfaces are produced at a resolution of 1x1 km for all methods examined in the study. The 'Out of range' label indicates predicted values that are outside the possible range of values of the indicator (below 0 or above 1). Out-of-range predictions were made by thin plate spline and universal kriging. National boundaries were downloaded from GADM.
(TIF)

**S10 Fig. Uncertainty maps of the predicted ITN ownership.** Uncertainty is measured as a prediction variance (var) for kriging methods (a, b) and as a standard deviation (SD) for Bayesian models (c). Higher values of SD or variance indicate areas with greater uncertainty in the predicted indicator, reflecting lower confidence in the accuracy of the predictions in these regions. ITN stands for insecticide-treated net. National boundaries were downloaded from GADM.
(TIF)

**S11 Fig. Predicted ITN ownership for 2 at 1x1 km for Senegal with each modelling approach.** The maps show the spatial distribution of the proportion (ranging from 0 to 1) of households with at least one insecticide-treated net (ITN) for every two people who slept in the house the night before the survey. Gridded surfaces are produced at a resolution of 1x1 km for all methods examined in the study. The 'Out of range' label indicates predicted values that are outside the possible range of values of the indicator (below 0 or above 1). Out-of-range predictions were made by universal kriging. National boundaries were downloaded from GADM.
(TIF)

**S12 Fig. Uncertainty maps of the predicted ITN ownership for 2.** Uncertainty is measured as a prediction variance (var) for kriging methods (a, b) and as a standard deviation (SD) for Bayesian models (c). Higher values of SD or variance indicate areas with greater uncertainty in the predicted indicator, reflecting lower confidence in the accuracy of the predictions in these regions. ITN stands for insecticide-treated net. National boundaries were downloaded from GADM.
(TIF)

**S13 Fig. Predicted ITN access at 1x1 km for Senegal with each modelling approach.** The maps show the spatial distribution of the proportion (ranging from 0 to 1) of population with access to an insecticide-treated net (ITN) in their

household in Senegal. Gridded surfaces are produced at a resolution of 1x1 km for all methods examined in the study. National boundaries were downloaded from GADM.
(TIF)

**S14 Fig. Uncertainty maps of the predicted ITN access.** Uncertainty is measured as a prediction variance (var) for kriging methods (a, b) and as a standard deviation (SD) for Bayesian models (c). Higher values of SD or variance indicate areas with greater uncertainty in the predicted indicator, reflecting lower confidence in the accuracy of the predictions in these regions. ITN stands for insecticide-treated net. National boundaries were downloaded from GADM.
(TIF)

**S15 Fig. Predicted IRS coverage at 1x1 km for Senegal with each modelling approach.** The maps show the spatial distribution of the proportion (ranging from 0 to 1) of households that were sprayed with a residual insecticide in the last year prior to the survey. Gridded surfaces are produced at a resolution of 1x1 km for all methods examined in the study. The 'Out of range' label indicates predicted values that are outside the possible range of values of the indicator (below 0 or above 1). Out-of-range predictions were made by thin plate spline, ordinary kriging and universal kriging. IRS stands for indoor residual spraying. National boundaries were downloaded from GADM.
(TIF)

**S16 Fig. Uncertainty maps of the predicted IRS coverage.** Uncertainty is measured as a prediction variance (var) for kriging methods (a, b) and as a standard deviation (SD) for Bayesian models (c). Higher values of SD or variance indicate areas with greater uncertainty in the predicted indicator, reflecting lower confidence in the accuracy of the predictions in these regions. IRS stands for indoor residual spraying. National boundaries were downloaded from GADM.
(TIF)

**S17 Fig. Predicted proportion of Fula people at 1x1 km for Senegal with each modelling approach.** The maps show the spatial distribution of the proportion (ranging from 0 to 1) of people belonging to the Fula ethnic group in Senegal. Gridded surfaces are produced at a resolution of 1x1 km for all methods examined in the study. The 'Out of range' label indicates predicted values that are outside the possible range of values of the indicator (below 0 or above 1). Out-of-range predictions were made by thin plate spline and universal kriging. National boundaries were downloaded from GADM.
(TIF)

**S18 Fig. Uncertainty maps of the predicted proportion of Fula people.** Uncertainty is measured as a prediction variance (var) for kriging methods (a, b) and as a standard deviation (SD) for Bayesian models (c). Higher values of SD or variance indicate areas with greater uncertainty in the predicted indicator, reflecting lower confidence in the accuracy of the predictions in these regions. National boundaries were downloaded from GADM.
(TIF)

## Author contributions

**Conceptualization:** Camille Morlighem, Catherine Linard.

**Formal analysis:** Camille Morlighem, Chibuzor Christopher Nnanatu, Atoumane Fall, Catherine Linard.

**Investigation:** Camille Morlighem.

**Methodology:** Camille Morlighem, Chibuzor Christopher Nnanatu, Corentin Visée.

**Project administration:** Catherine Linard.

**Resources:** Chibuzor Christopher Nnanatu, Atoumane Fall.

**Software:** Camille Morlighem, Chibuzor Christopher Nnanatu, Corentin Visée.

**Supervision:** Chibuzor Christopher Nnanatu, Catherine Linard.

**Writing – original draft:** Camille Morlighem.

**Writing – review & editing:** Camille Morlighem, Chibuzor Christopher Nnanatu, Corentin Visée, Atoumane Fall, Catherine Linard.

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
