## [Decision Letter · Decision Letter 0]

15 Dec 2024

PONE-D-24-15006Spatial interpolation of health and demographic variables: predicting malaria indicators with and without covariatesPLOS ONE

Dear Dr. Morlighem,

Thank you for submitting your manuscript to PLOS ONE. After careful consideration, we feel that it has merit but does not fully meet PLOS ONE’s publication criteria as it currently stands. Therefore, we invite you to submit a revised version of the manuscript that addresses the points raised during the review process.

We look forward to receiving your revised manuscript.

Kind regards,

Jinyi Wu, MD

Academic Editor

PLOS ONE

Journal Requirements:

2. We note that Figures 3,4 and S1-S16 in your submission contain [map/satellite] images which may be copyrighted. All PLOS content is published under the Creative Commons Attribution License (CC BY 4.0), which means that the manuscript, images, and Supporting Information files will be freely available online, and any third party is permitted to access, download, copy, distribute, and use these materials in any way, even commercially, with proper attribution. For these reasons, we cannot publish previously copyrighted maps or satellite images created using proprietary data, such as Google software (Google Maps, Street View, and Earth). For more information, see our copyright guidelines: http://journals.plos.org/plosone/s/licenses-and-copyright.

a. You may seek permission from the original copyright holder of Figures 3,4 and S1-S16 to publish the content specifically under the CC BY 4.0 license. 

Reviewers' comments:

Reviewer's Responses to Questions

**Comments to the Author**

1. Is the manuscript technically sound, and do the data support the conclusions?

Reviewer #1: Yes

Reviewer #2: Yes

2. Has the statistical analysis been performed appropriately and rigorously? 

Reviewer #1: Yes

Reviewer #2: Yes

3. Have the authors made all data underlying the findings in their manuscript fully available?

Reviewer #1: Yes

Reviewer #2: Yes

4. Is the manuscript presented in an intelligible fashion and written in standard English?

Reviewer #1: Yes

Reviewer #2: Yes

5. Review Comments to the Author

Reviewer #1: The title of the paper by Morlighem et al., “Spatial interpolation of health and demographic variables: predicting malaria indicators with and without covariates”, and in general, I like the idea. However, I have some concerns about the lack of details in the methodology.

1) What is the primary aim of this study? I can only see to compare three categories of methods for predicting DHS indicators. What are the sub-objectives?

2) You have missed an opportunity to include others socio-demographic factor such as religious beliefs, can have a direct impact on human health. More specifically, a sociological factor may lead to parents objecting to immunization on religious or philosophical grounds. Access to medical care, GDP, seasonality, humidity, population density and migrations are also important risk factors.

3) I realize that much of the methods used are quite technical, but it is quite hard to understand what the authors did via the main body of the text and some of the more complex methods should be described here rather than simply indicating these to the appendix.

4) A more thorough description of what spline were used to describe the association beyond simply stating that a more flexible model might be needed. I would also like to have seen a table with the AICs, BICs and residuals in the appendix to justify model choice.

5) I am also wondering why a model considering the interplay between all parameters was never considered in these analyses. It also seems like this would be fairly easy to do given this model framework and it would be useful to know why this was excluded.

5) I am curious as to why authors did not plot their findings using gradient mapping for all main outcomes. This would easily be done and would make the findings more compelling for readers. Since authors have already performed the necessary steps in performing Bayesian kringing, it would suffice to simply plot contour maps using existing packages in R (ex: geoR package, spbayes etc) or doing so using Stan (RStan). It seems a shame not to take the analysis to its logical conclusion and present the findings in this way.

Reviewer #2: Thank you for submitting this well-structured manuscript, which provides a robust analysis of geospatial modeling approaches to predict health and development indicators. The manuscript demonstrates a high level of technical soundness, with a well-designed comparison of Bayesian geostatistical models, random forests, and spatial interpolation techniques. The use of evaluation metrics such as pseudo-R², RMSE, and MAE effectively supports the conclusions, and the statistical methods are appropriately chosen and rigorously applied. However, the discussion would benefit from a deeper exploration of the assumptions and potential limitations of each modeling approach, as well as the generalizability of the findings beyond the Senegal context and the specific DHS indicators analyzed.

The statistical analysis is thorough and rigorous, with a robust comparison of spatial and non-spatial models. The study benefits from the use of cross-validation and comprehensive model evaluation metrics. However, greater transparency in the cross-validation procedures, such as the number of folds or randomization methods, would enhance reproducibility. Similarly, a more detailed explanation of predictor selection processes, including how multicollinearity was addressed, would strengthen the analysis. Additionally, discussing potential biases from missing data or the spatial distribution of survey points would provide a more complete understanding of the results.

The Data Availability Statement aligns with PLOS data-sharing requirements, noting that DHS datasets are publicly accessible upon registration and approval. While this satisfies the policy, the authors should ensure that the exact datasets used (e.g., year, country, and modules) are clearly specified. Sharing any additional data products, such as interpolated predictions or model outputs, in a public repository or supplementary materials would further enhance reproducibility. Including scripts or code for data processing and analysis is also encouraged to facilitate transparency.

The manuscript is well-written and presented in standard English, but minor revisions could improve readability. Some overly complex sentences, particularly in the methods and results sections, could be simplified for greater clarity. Abbreviations such as DHS, RMSE, and MAE should be defined at first mention to ensure accessibility for a broader audience. The flow of the discussion section could be improved by systematically organizing the key findings and their implications, which would make the narrative more engaging and easier to follow.

This study contributes valuable insights into the application of geospatial models to health and development indicators. The comparative analysis highlights the strengths and weaknesses of different approaches, offering practical implications for future modeling efforts. However, expanding on the broader applicability of the findings to other contexts and indicators would enhance the manuscript’s impact. Additionally, discussing how the results could inform policymaking and addressing any ethical considerations related to geospatial data usage would strengthen the overall narrative.

Lastly, the inclusion of visualizations such as predicted indicator maps would make the findings more tangible and engaging for readers. Ensuring that all figures and tables are referenced in the correct order and that captions are sufficiently detailed for standalone interpretation would further improve the manuscript’s presentation. While no significant ethical concerns are apparent, a brief mention of privacy and data security considerations when handling spatial data would provide additional reassurance.

Overall, the manuscript demonstrates high technical quality and relevance to the field. Addressing the points above would further enhance its clarity, rigor, and broader applicability. This is a valuable contribution that merits consideration for publication after revisions.

6. PLOS authors have the option to publish the peer review history of their article (what does this mean? ). If published, this will include your full peer review and any attached files.

**Do you want your identity to be public for this peer review?** For information about this choice, including consent withdrawal, please see our Privacy Policy .

Reviewer #1: No

Reviewer #2: No

---

## [Author Response · Author response to Decision Letter 1]

22 Jan 2025

We would like to thank the Editor and the reviewers for their comments and thorough reviews of this manuscript. We hope to have satisfactorily addressed most of their comments in the revised version and we have provided a detailed answer to each of them. In addition, the manuscript has been proofread for spelling mistakes and to simplify complex sentences.

Journal Requirements:

The manuscript has been checked for compliance with PLOS ONE’s style requirements.

2) We note that Figures 3,4 and S1-S16 in your submission contain [map/satellite] images which may be copyrighted. All PLOS content is published under the Creative Commons Attribution License (CC BY 4.0), which means that the manuscript, images, and Supporting Information files will be freely available online, and any third party is permitted to access, download, copy, distribute, and use these materials in any way, even commercially, with proper attribution. For these reasons, we cannot publish previously copyrighted maps or satellite images created using proprietary data, such as Google software (Google Maps, Street View, and Earth). For more information, see our copyright guidelines: http://journals.plos.org/plosone/s/licenses-and-copyright.

The copyrights for Figures 3–5 and S1–S18 are held by the authors of this manuscript, as the continuous surfaces displayed on the maps were generated by the authors and represent original results from this study.

The national boundaries of Senegal and neighbouring countries were downloaded from the GADM website (https://gadm.org/), which states that the data are free for use in scientific papers: “The data are freely available for academic use and other non-commercial use. Redistribution or commercial use is not allowed without prior permission. Using the data to create maps for publishing of academic research articles is allowed. Thus you can use the maps you made with GADM data for figures in articles published by PLoS, Springer Nature, Elsevier, MDPI, etc. You are allowed (but not required) to publish these articles (and the maps they contain) under an open license such as CC-BY as is the case with PLoS journals and may be the case with other open access articles.” (see: https://gadm.org/license.html#:~:text=The%20data%20are%20freely%20available,not%20allowed%20without%20prior%20permission). In addition, we have credited GADM data provider under all figures using GADM data.

Reviewer 1: The title of the paper by Morlighem et al., “Spatial interpolation of health and demographic variables: predicting malaria indicators with and without covariates”, and in general, I like the idea. However, I have some concerns about the lack of details in the methodology.

Thank you, we are pleased to read that you like the idea of the paper. We hope that we have addressed your concerns in the revised version and in our point-by-point responses below.

1) What is the primary aim of this study? I can only see to compare three categories of methods for predicting DHS indicators. What are the sub-objectives?

Thank you for your comment. As you point out, the manuscript needed some clarification regarding the main objectives.

The aim of this paper is to produce continuous surfaces of useful malaria-related indicators from the DHS with the following sub-objectives: 1) compare three categories of methods for predicting DHS indicators (spatial interpolation methods, ensemble methods and Bayesian geostatistical models), 2) assess the added value of covariate-based methods over methods that rely only on spatial autocorrelation, and 3) provide a comprehensive assessment of the strengths and weaknesses of these methods to guide users in their choice. This has been clarified in the introduction of the manuscript.

2) You have missed an opportunity to include others socio-demographic factor such as religious beliefs, can have a direct impact on human health. More specifically, a sociological factor may lead to parents objecting to immunization on religious or philosophical grounds. Access to medical care, GDP, seasonality, humidity, population density and migrations are also important risk factors.

This is a very good point! Religious and cultural beliefs can indeed influence health prevention and treatment behaviours. However, in Senegal, religion is relatively homogeneous, with more than 95% of the population being Muslim and the rest Christian (2017 and 2023 DHS estimates), which may result in less variability in prevention behaviours influenced by religion. In contrast, there is significant ethnic diversity, and cultural beliefs associated with ethnicity may shape health perceptions and practices. In addition, ethnicity may influence genetic immunity to malaria.

To account for this, we included an indicator representing the prevalence of Fula people - an ethnic group known to have a lower susceptibility to malaria due to genetic factors. Some Fula groups in Senegal are also nomadic, which may influence cultural beliefs and behaviours related to health due to their migratory lifestyle. Model results for this indicator have been added to the results section, and we have also highlighted in the introduction that “ethno-religious beliefs may also influence perceptions of malaria and the uptake of preventive measures”. Thank you for this suggestion, which has strengthened the conclusions of the paper. Fula ethnicity was better predicted using covariate-independent methods, as were the variables relating to malaria prevention. This highlights that variables with a heterogeneous spatial structure tend to be better modelled with models that rely on spatial autocorrelation alone and do not need covariates. This is discussed in more detail in the discussion (third paragraph).

As for the other factors you mention, they are indeed relevant risk factors. However, variables such as humidity, seasonality, population density and GDP cannot be derived from the DHS data and were therefore not the focus of this study, especially as these are already covered by freely available data sources (WorldPop, WorldClim, etc.). Access to health care has been the focus of previous work [Bihin et al., 2022] which compared accessibility methods in different settings. Migration (although indirectly considered with the indicator on Fula ethnicity) and access to health care are now mentioned in the discussion as a potential perspective for extending the analysis to other indicators: “We encourage future research to replicate the methods of this study to other countries and to additional malaria-related indicators, such as access to health care and the prevalence of migrants, to further validate and extend these findings.”

Bihin J, De Longueville F, Linard C. Spatial accessibility to health facilities in Sub-Saharan Africa: comparing existing models with survey-based perceived accessibility. Int J Health Geogr. 2022 Nov 12;21(1):18.

3) I realize that much of the methods used are quite technical, but it is quite hard to understand what the authors did via the main body of the text and some of the more complex methods should be described here rather than simply indicating these to the appendix.

Thank you for your comment. We understand that the methods section lacked sufficient detail to fully understand how the methods work. To improve clarity, the description of the methods has been moved from the appendix to the main text, except for Kriging and Bayesian models where less important details were kept in the appendix to manage the length of the method section.

4) A more thorough description of what spline were used to describe the association beyond simply stating that a more flexible model might be needed. I would also like to have seen a table with the AICs, BICs and residuals in the appendix to justify model choice.

Splines are more flexible methods in the sense that the interpolated surface can pass exactly through the sample points or can deviate slightly from them, unlike typical interpolation methods. Following your comment, this has been clarified in the methods section. In the discussion section, we explained further why splines work well for intervention-related indicators (instead of simply saying that a flexible model is needed): “... Access to malaria prevention may be influenced by policy decisions on intervention planning, resulting in strong spatial autocorrelation in areas (e.g. administrative unit, health district) where interventions are implemented. This may explain why TPS performed well for these indicators. The interpolated surface can adapt to both abrupt changes, capturing local variations (by passing through the sample points), and smooth trends (by deviating from the sample points) [20,60]. This makes TPS particularly suitable for datasets with heterogeneous spatial structures (e.g. variable values change abruptly in space, as seen with intervention-related variables and ethnic groups)”.

As for the additional metrics, we did not include AIC and BIC values because these metrics are not applicable to methods such as Inverse Distance Weighting (IDW) and Random Forest models. These metrics are based on log-likelihood, which is not explicitly available for these models. Instead, we have added the residuals from the model fit (measured as a mean absolute error) in the appendix (Table S1), as you suggested. Note, however, that these were not used to justify model choice, as they are not directly comparable between models based on different assumptions. For example, residuals are zero for IDW because the interpolant is exact (it passes through the sample points), but residuals are higher for splines because the surface can deliberately deviate from the sample points. As they were not directly comparable, we compared the methods based on their performance in cross-validation.

5) I am also wondering why a model considering the interplay between all parameters was never considered in these analyses. It also seems like this would be fairly easy to do given this model framework and it would be useful to know why this was excluded.

Good point! Bayesian and kriging models with interactions between covariates were not implemented in this study for several reasons: (1) multicollinearity would be more difficult to control (2) model complexity would increase and (3) computational complexity would significantly increase given the large numbers of covariates. Following your comment, we have added in the discussion that considering interactions between covariates could improve the models of this study. Note that random forest models inherently allow for interactions between covariates, as they are based on decision trees (this has also been added to the section where methods are compared).

Similarly, models with multiple DHS outcomes were not used for reasons of model and computational complexity. Instead, we focused on comparing how the methods performed for different indicators separately. However, considering the correlation between all model outcomes is highly relevant, and those focusing on multiple outcomes could consider building a joint Bayesian model if interpretation and computational efficiency are not a primary concern. We have now added that joint modelling may be a suitable option for modelling multiple outcomes in the discussion: “Research that aims to model multiple outcomes simultaneously could investigate joint Bayesian models, which account for correlations between the spatial structures of all response variables. However, these models are significantly more complex and require more computational resources”.

6) I am curious as to why authors did not plot their findings using gradient mapping for all main outcomes. This would easily be done and would make the findings more compelling for readers. Since authors have already performed the necessary steps in performing Bayesian kringing, it would suffice to simply plot contour maps using existing packages in R (ex: geoR package, spbayes etc) or doing so using Stan (RStan). It seems a shame not to take the analysis to its logical conclusion and present the findings in this way.

Thank you for your suggestion, which was also raised by the second reviewer. We initially did not include all figures (10 in total) in the manuscript to avoid overloading the paper with visuals, but we agreed that it was a missed opportunity not to show maps for all outcomes. We have therefore added a predictive map for all indicators (see Figure 3). Each outcome is mapped using the best performing method. We also changed the colour scale to a colour-blind friendly one, which is also more intuitive than the previous one (clear colours now indicate low outcome values).

Reviewer 2: Thank you for submitting this well-structured manuscript, which provides a robust analysis of geospatial modeling approaches to predict health and development indicators. The manuscript demonstrates a high level of technical soundness, with a well-designed comparison of Bayesian geostatistical models, random forests, and spatial interpolation techniques. The use of evaluation metrics such as pseudo-R², RMSE, and MAE effectively supports the conclusions, and the statistical methods are appropriately chosen and rigorously applied.

Thank you very much for your comment. We are pleased to read that you find merit in our work.

1) However, the discussion would benefit from a deeper exploration of the assumptions and potential limitations of each modeling approach, as well as the generalizability of the findings beyond the Senegal context and the specific DHS indicators analyzed.

Great suggestions! We now summarise the strengths and weaknesses of the methods explored in this study in the discussion (see fourth paragraph). As you pointed out, the manuscript lacked generalisation of the findings to other settings. Instead, it stated that these findings were limited to the data used, when in fact they may be applicable in other contexts. We have therefore revised the discussion and added a paragraph to address this: “Although we focused only on indicators from the Senegal 2017 DHS, findings of this study may be useful in other settings. The results may be generalisable to other DHS surveys, as these are conducted according to standardised protocols and are comparable across countries [10]. In addition, we found that methods based on spatial autocorrelation alone performed better for malaria prevention and ethnicity variables due to their heterogeneous spatial structures. This is likely to be true for other variables with similar structures (e.g. vaccination coverage, which also depends on policy decisions, or prevalence of female genital mutilation, which is related to ethno-cultural factors [23]). Lastly, the strengths, weaknesses and recommendations regarding the methods used in this study, along with the criteria for choosing a method and the codes provided with this paper, are relevant for modelling other DHS indicators and demographic and health variables in general. We encourage future research to replicate the methods of this study to other countries and to additional malaria-related indicators, such as access to health care and the prevalence of migrants, to further validate and extend these findings.”

2) The statistical analysis is thorough and rigorous, with a robust comparison of spatial and non-spatial models. The study benefits from the use of cross-validation and comprehensive model evaluation metrics. However, greater transparency in the cross-validation procedures, such as the number of folds or randomization methods, would enhance reproducibility. Similarly, a more detailed explanation of predictor selection processes, including how multicollinearity was addressed, would strengthen the analysis. Additionally, discussing potential biases from missing data or the spatial distribution of survey points would provide a

---

## [Decision Letter · Decision Letter 1]

28 Mar 2025

Spatial interpolation of health and demographic variables: predicting malaria indicators with and without covariates

PONE-D-24-15006R1

Dear Dr. Morlighem,

We’re pleased to inform you that your manuscript has been judged scientifically suitable for publication and will be formally accepted for publication once it meets all outstanding technical requirements.

Kind regards,

Jinyi Wu, MD

Academic Editor

PLOS ONE

Additional Editor Comments (optional):

Reviewers' comments:

Reviewer's Responses to Questions

**Comments to the Author**

1. If the authors have adequately addressed your comments raised in a previous round of review and you feel that this manuscript is now acceptable for publication, you may indicate that here to bypass the “Comments to the Author” section, enter your conflict of interest statement in the “Confidential to Editor” section, and submit your "Accept" recommendation.

Reviewer #1: All comments have been addressed

Reviewer #2: All comments have been addressed

2. Is the manuscript technically sound, and do the data support the conclusions?

Reviewer #1: Yes

Reviewer #2: Yes

3. Has the statistical analysis been performed appropriately and rigorously? 

Reviewer #1: Yes

Reviewer #2: Yes

4. Have the authors made all data underlying the findings in their manuscript fully available?

Reviewer #1: Yes

Reviewer #2: Yes

5. Is the manuscript presented in an intelligible fashion and written in standard English?

Reviewer #1: Yes

Reviewer #2: Yes

6. Review Comments to the Author

Reviewer #1: I have reviewed the author's responses and am satisfied that all my concerns have been addressed. I have no further comments.

Reviewer #2: Thank you for your thorough and well-structured revision. The manuscript has been significantly improved, with enhanced methodological clarity, transparency, and depth of discussion. The responses to reviewer comments demonstrate careful attention to detail and a commitment to ensuring the rigor and reproducibility of the analysis.

7. PLOS authors have the option to publish the peer review history of their article (what does this mean? ). If published, this will include your full peer review and any attached files.

**Do you want your identity to be public for this peer review?** For information about this choice, including consent withdrawal, please see our Privacy Policy .

Reviewer #1: No

Reviewer #2: No

---

## [Editor Report · Acceptance letter]

PONE-D-24-15006R1

PLOS ONE

Dear Dr. Morlighem,

I'm pleased to inform you that your manuscript has been deemed suitable for publication in PLOS ONE. Congratulations! Your manuscript is now being handed over to our production team.

Kind regards,

on behalf of

Dr. Jinyi Wu

Academic Editor

PLOS ONE